# Perceived Information Overload and Intention to Discontinue Use of Short-Form Video: The Mediating Roles of Cognitive and Psychological Factors

**DOI:** 10.3390/bs13010050

**Published:** 2023-01-06

**Authors:** Donghwa Chung, Yuanxin Chen, Yanfang Meng

**Affiliations:** 1School of Journalism and Communication, Shanghai University, Shanghai 200444, China; 2School of Journalism and Communication, Beijing Institute of Graphic Communication, Beijing 102699, China; 3Network and New Media, Beijing Institute of Graphic Communication, Beijing 102699, China

**Keywords:** perceived information overload, discontinuous usage, short-form video, stress-coping theory

## Abstract

The current study investigated the effects of Chinese young adult users’ perceived information overload (i.e., the daily perception of exposure to excessive information) on their intention to stop using short-form video applications. Specifically, this study accomplished this by measuring the direct and indirect effects of social media fatigue, maladaptive coping, and life dissatisfaction in relation to users’ intention to discontinue their use of short-form video applications. The data were collected using a web-based survey and validated questionnaire, with a sample of 340 young adult (18–26 years old) respondents. The results indicated that perceived information overload had a direct effect on the intention to discontinue the use of short-form video applications. Moreover, short-form video fatigue, maladaptive coping, and life dissatisfaction all played mediating roles in the relationship between perceived information overload and the intention to discontinue the use of short-form video applications among young adults in China.

## 1. Introduction

In recent years, the influence of social media has risen continuously [1]. Despite the growing popularity of Facebook and WeChat, there is evidence that the number of daily visits and the total time spent on Facebook and other SNSs (social networking sites) by users has reduced drastically [2]. Similarly, Chinese short-form video applications have been shown to be susceptible to declining user interest during the COVID-19 pandemic [3]. For instance, the average daily active users of Kuaishou (the second-largest short-form video service in China) decreased by 2.1 million in the second quarter of 2021, and the average monthly active users fell by 13.6 million [4]. Additionally, since January 2018, the number of newly added users has declined as the Douyin, Kuaishou, Huaoxiao, and Watermelon video applications have reached mature industry status [5]. However, the mechanisms behind such phenomena during the pandemic in China remain underexplored.

In recent years, short-form video applications have been recognized as the fastest-growing forms of social media [6]. They are also known to be among the leading entertainment and social media applications in China [7]. Specifically, Douyin (known as TikTok overseas) has reached 600 million daily active users [8]. Short-form video applications fit in well with Chinese young adult users’ busy lives, provide entertainment, and have been applied to every aspect of Chinese peoples’ daily lives [9]. Regardless of its growing popularity and advantages, the dark side of social media usage has attracted the attention of researchers in recent years [10]. For instance, Liu et al. [11] found that gen Z social media users’ social media fatigue and fear of COVID-19 have increased their willingness to discontinue the use of social media. The discontinued use of social media is recognized as one of the negative outcomes of problematic media use [11,12,13]. This has amplified the effect of losing users from social media platforms, which in turn may be harmful to the growth of the social media industry [14]. Users quitting or switching applications costs users time and effort to adapt to new social media platforms [13,15]. Eventually, social media providers may lose their base users as well as potential revenue [16]. As discussed above, investigating this phenomenon is critically important for two reasons. First, short-form video application developers are eager to understand why users have tended to discontinue the use of their products during the COVID-19 pandemic [17]. Second, no empirical studies have applied a theoretical framework or identified the mechanisms that lead to the discontinued use of short-form applications.

The discontinued use of media has been discussed thoroughly in the field of digital and social media [18,19]. Fu et al. [12] conceptualized the discontinued use of media as users making the decision to reduce or abandon their use of social media. Previous research has demonstrated that the perception of exposure to excessive amounts of information has led to individuals discontinuing social media use or switching social media platforms during the COVID-19 pandemic [11,13]. Stress-coping theory indicates that maladaptive coping strategies involve individuals’ behavioral attempts to decrease their level of discomfort when they are maintaining negative emotional states [20,21]. Based on this theory, a previous study found that maladaptive coping is positively associated with individuals’ discontinued use of social media [22]. Moreover, Rajkumar [23] indicated that perceived threats related to COVID-19 have not only reduced the quality of individuals’ physical health but also their life satisfaction levels. Additionally, problematic internet use is recognized as one of the factors that has decreased Palestinians’ life satisfaction during the COVID-19 pandemic [21]. Ultimately, this has amplified individuals’ discontinued use of media [24,25]. Overall, prior studies have provided the clear and convincing finding that cognitive (short-form video fatigue and maladaptive coping) and psychological factors (life dissatisfaction) have increased individuals’ discontinuation of the use of social media. However, to the best of our knowledge, whether these factors interact to produce the intention to discontinue the use of short-form video applications amid the COVID-19 pandemic in China remains largely underexplored.

To fill this gap, this study drew on stress-coping theory as a guide, further exploring the effect of the perceived information overload from short-form video applications on Chinese young adult (18–26 years old) users’ intent to stop using said applications. This study also explored the mediating roles of short-form video fatigue, maladaptive coping, and life dissatisfaction in this process.

## 2. Literature Review

### 2.1. Stress-Coping Theory and Maladaptive Coping

Lazarus and Folkman defined coping as an individual’s response to stressful situations involving important and potentially negative consequences [26]. Several scholars have developed different types of coping categories [27,28]. Among those categories, two dimensions of coping categories have been widely applied in prior studies [29,30,31]. A recent study defined adaptive coping as a strategy that gives an individual the time he/she needs to self-reflect and refocus on what is important, whereas maladaptive coping refers to a strategy that involves efforts to regulate the psychological consequences of stressful or potentially stressful events [32]. However, maladaptive strategies deny the user feasible measures to improve the situation and encourage them to use their energy to suppress thoughts, emotions, and actions related to coping with stress, which results in users feeling that their ability to cope is increasingly limited [22]. In social media use and consumer behavioral studies, coping theory has been explained as the way individuals adapt to and handle social media overload. For instance, a previous study demonstrated that while users are dealing with technostressors, maladaptive strategies are likely to be used, which has unhealthy outcomes for perceiving stress [15]. Additionally, individuals who rely on maladaptive coping strategies tend to have emotional and negative consequences (e.g., anger, avoidance, or disengaged coping) [33]. In the context of the COVID-19 pandemic in China, Ni et al. [34] indicated that Chinese adults have spent too much time on social media searching for COVID-19-related information updates, which in turn has further increased their symptoms of depression and anxiety. Furthermore, a previous study demonstrated that perceived stress amid the COVID-19 pandemic has worsened Chinese users’ level of negative coping [35]. In response to this issue, individuals are more likely to use maladaptive coping strategies to deal with perceived technostressors as opposed to applying adaptive coping strategies, such as avoiding the use of social media. For instance, Lin [22] suggested that the more likely individuals are to use maladaptive coping styles, the more likely they are to discontinue their media usage. Considering the studies discussed above, maladaptive coping strategies can be applied to the intention to discontinue the use of short-form video applications in China and provide a clarifying perspective for the context of this research.

### 2.2. Effect of Information Overload on Intention and Cognitive and Psychological Factors

In recent years, cutting-edge new media platforms have greatly attracted Chinese young adults’ interest. Short-form video applications are among the most frequently used communication platforms in China [7]. The content created on short-form video applications is primarily user-generated [11]. With easy-to-use operation interfaces and functional editing features, short-form video applications offer users a variety of sound effects, video filters, and the ability to upload short-form videos directly from their phone [36,37]. As the latest Chinese investigation demonstrated, the majority of Douyin (TikTok) users are interested in short-form videos on topics such as singing, cooking, physical exercise, traveling, comedy, lip-syncing, and knowledge sharing (e.g., scientific knowledge, health-related knowledge, and skills) [38]. Although positive usage behaviors in short-form video applications have previously been demonstrated [9], scholars have recently focused on investigating whether information overload on TikTok could lead to negative outcomes, including the discontinued use of short-form video applications [39], short-form video fatigue [39], maladaptive coping [40], and life dissatisfaction [41]. Undoubtedly, these studies have suggested that information overload on TikTok directly influences individuals’ cognitive and psychological wellbeing. However, previous studies have failed to investigate such mechanisms during the COVID-19 pandemic in China. They have also failed to identify how information overload from Chinese short-form video applications leads to negative outcomes. Therefore, further investigation is required.

Farhoomand and Drury [42] claimed that when an individual perceives themselves as being exposed to excessive amounts of information, this impairs their ability to handle that information. This phenomenon is conceptualized as “information overload.” In most cases, information overload causes media users to feel overburdened and to experience an increasing level of stress, which results in a negative emotion arousal effect [11]. This study conceptualizes perceived information overload as a phenomenon whereby users receive too much information and communication in a short period of time from push notifications and recommended videos from short-form video applications. A previous study demonstrated that there was a positive relationship between individuals’ perceived information overload and Facebook users’ discontinued usage of the platform [12]. Similarly, a recent study found that social media users’ perceived information overload on Weibo was positively associated with their discontinued usage of the platform [43]. Therefore, Chinese young adult users’ information overload on short-form video applications is likely to increase their intention to discontinue their use of short-form video applications.

One of the critical factors that predicts individuals’ discontinued use of social media is social media fatigue [39]. Social media fatigue is described as a decrease in individuals’ interest in using social media [44]. In the context of this research, short-form video fatigue refers to the decrease in users’ interest in using those applications or to users’ intention to stop using short-form video applications. Cyberpsychology researchers have also investigated how perceived information overload can have an impact on how social media users deal with perceived technostress and suffer from feelings of dissatisfaction. For instance, when social media users experienced perceived information overload on social networks, it induced platform fatigue on these networks [45]. Moreover, one scholar found that individuals’ perception of being overloaded with information about COVID-19 was positively associated with their level of social media fatigue [11]. Likewise, a previous study also demonstrated that there is a positive relationship between WeChat users’ perceived information overload and their social media fatigue [46]. Thus, as Chinese young adult users’ information overload on short-form video applications increases, eventually this will increase their level of fatigue with short-form video applications.

Two types of coping strategies are involved in efforts to release stress from various stressful events. Among these coping strategies, maladaptive coping strategies were largely applied by Chinese social media users when they experienced information overload [22]. In our study, maladaptive coping refers to how individuals deal with perceived technostress from short-form videos by adopting a negative solution (e.g., avoidance or anger) and suffering from feelings of dissatisfaction. Recent studies have investigated whether individuals’ perceived information overload amplified their use of maladaptive coping strategies [22,47]. Media behavior scholars have found that when social media users are exposed to excessive information on WeChat, their level of maladaptive coping (e.g., giving up on dealing with stressful situations on WeChat and efforts to cope with exhaustion) increases [48]. Likewise, when college students were exposed to excessive amounts of health information online, instead of choosing positive ways to cope with perceived technostress, they expressed feelings of anger [49]. Therefore, the current study predicted that perceived information overload on short-form applications would be associated an increased level of maladaptive coping.

The factors that determine an individuals’ life dissatisfaction include health behavior, social factors, personality, and mood factors [26]. The current study conceptualizes life dissatisfaction as a reflection of individuals’ negative self-evaluation of their quality of life, according to their own perceptions, after being exposed to an overload of information on short-form video applications. Previously, Alheneidi [50] explored users’ excessive engagement in social media and how being exposed to information overload leads to an increase in their life dissatisfaction (e.g., distress, negative feelings from social comparison, and negative wellbeing). Perceived COVID-19 information overload is a strong predictor of individuals’ negative wellbeing (life dissatisfaction) [44]. To sum up, it is logical that Chinese young adults’ perceived information overload on short-form video applications may increase their level of life dissatisfaction. The following hypothesis was thus proposed:

**H1.** 
*Perceived information overload has a significant and positive effect on (a) the intention to discontinue the use of short-form video applications, (b) short-form video fatigue, (c) maladaptive coping, and (d) life dissatisfaction.*


### 2.3. The Roles of Cognitive Factors as Mediators

A handful of researchers have found that there is a positive relationship between individuals’ perceived information overload and their levels of social media fatigue [11,27,45] Furthermore, the greater the media fatigue, the more likely it is that users will discontinue their use of social media [10,51]. Similarly, a Chinese study demonstrated that social media users’ exposure to excessive amounts of COVID-19-related information amplified their level of social media fatigue [11], which in turn may increase users’ intention to stop using social media [11,52].

However, previous studies related to media and behavior have demonstrated that social media users’ perceived information overload increases their level of maladaptive coping [22,53,54]. Furthermore, this has further reinforced individuals’ discontinuation of the use of social media [22]. To sum up, it is logical that Chinese young adults’ perceived information overload would increase their level of maladaptive coping. As a result, this would further affect their discontinued use of short-form video applications. The following hypotheses were thus proposed:

**H2a.** 
*Short-form video fatigue mediates the relationship between perceived information overload and the intention to discontinue the use of short-form video applications.*


**H2b.** 
*Maladaptive coping mediates the relationship between perceived information overload and the intention to discontinue the use of short-form video applications.*


### 2.4. The Roles of Psychological Factors as a Mediators

Scholars have discussed users’ excessive engagement with social media and how being exposed to information overload leads to increases in their life dissatisfaction (e.g., distress, negative feelings from social comparison, and negative wellbeing) [50]. Eventually, this induces individuals to avoid the use of social media [24,25]. Moreover, exposure to COVID-19-related information has increased individuals’ levels of negative wellbeing (life dissatisfaction) [44], which are amplified along with their frequency of discontinuing the use of social media applications [47].

As mentioned above, previous literature has explored life dissatisfaction as one of the key psychological factors that indirectly affects this causal relationship. Therefore, this study argues that life dissatisfaction could mediate the effect of the relationship between perceived information overload and the intention to discontinue the use of short-form video applications. Hence, the following hypothesis was thus proposed (Figure 1):

**H3.** 
*Life dissatisfaction mediates the relationship between perceived information overload and the intention to discontinue the use of short-form video applications.*


## 3. Materials and Methods

### 3.1. Questionnaire Design

The current study adopted 5 measures (e.g., perceived information overload, media fatigue, maladaptive coping, life dissatisfaction, and discontinued usage) from existing studies. In addition, all questions related to each measurement were adapted from verified studies [2,22,24,55,56]. However, the measures had not previously been used in the context of Chinese culture. In order to maintain the precision and accuracy of the measures, this study carefully made appropriate modifications. First, the measures were translated from the original English into Chinese by three Chinese-speaking language experts. Later, all researchers compared the original and translated measures until they reached an agreement regarding the translation’s quality and accuracy [57]. Second, content and face validity methods were adopted to revise and eliminate items [58]. In this process, four experts from the Department of Journalism and Media were invited for the evaluation [57]. Third, a pretest was conducted with 15 volunteers who were recruited from Shanghai University and the Beijing Institute of Graphic Communication. Participants were asked to provide any recommendations for the revision if any item was unreadable or ambiguous [59]. We then revised the questionnaire based on the volunteers’ feedback and uploaded the finalized survey to the online survey platform Wenjuanxing. Wenjuanxing is also recognized as one of the most used survey design and dissemination platforms in China. In addition, the platform provides a sampling pool of nearly 260 million registered users in China (excluding Tibet). Based on these remarkable features, it has been utilized by numerous universities and scholars for conducting research related to China [60,61]. The users for this study were selected randomly, sent the survey link, and provided a reward upon their completion of the questionnaire. Permission to conduct the current study was reviewed and approved by the Beijing Institute of Graphic Communication Ethics Committee. The statement of permission was included in the beginning of the online questionnaire. The data were collected from 1 January to 1 October 2022 through an online survey platform. Among the 365 potential respondents who received the survey link, 340 filled out the questionnaire.

### 3.2. Measurement of Variables

#### 3.2.1. Perceived Information Overload

To observe perceived information overload, we derived the information overload scale from Misra and Stokols’s [55] study. This scale was used to evaluate individuals’ perception of whether they received too much information from short-form video applications. Respondents answered the following questions about their level of agreement using a 5-point Likert scale: (1) “It takes too much effort to manage my subscribed accounts lists’’; (2) “I often feel overwhelmed with the number of notifications I receive on short-form video platforms”; (3) “Sorting through all the information on short-form video platforms takes up too much of my time”; (4) I receive too many recommended videos on the applications (M = 2.80, SD = 2.80, α = 0.65).

#### 3.2.2. Short-Form Video Fatigue

The short-form video fatigue scale was adapted from Bright et al. [2]. The short-form video fatigue scale aimed to assess individuals’ decreased interest in using short-form video applications. Respondents answered regarding their level of agreement with the following questions using a 4-point Likert scale: (1) “I am not interested in all the new things that are happening on short-form video platforms”; (2) “After using short-form video applications, I feel tired or listless”; (3) “Short-form video applications make me feel very frustrated”; or (4) “After using short-form video applications, I feel extremely irritable” (M = 2.94, SD = 0.91, α = 0.84).

#### 3.2.3. Maladaptive Coping

The 3-item maladaptive coping scale was adapted from Lin et al. [22]. This scale evaluates how individuals cope with perceived technostress from short-form video applications by adopting a negative solution and suffering from unsatisfied feelings. Respondents provided their level of agreement to the following questions using a 3-point Likert scale: (1) “I refuse to believe that messy situations have happened on short-form video applications”, (2) “I have given up on trying to deal with stressful situations on short-form video applications”, or (3) “I have given up attempting to cope with stressful situations on short-form video applications” (M = 2.98, SD = 0.78, α = 0.79).

#### 3.2.4. Life Dissatisfaction

To measure life dissatisfaction, we adapted the life dissatisfaction scale created by Diener et al. [55]. The original scale was designed to measure individuals’ overall wellbeing. This study aimed to evaluate negative self-evaluations of quality of life after being exposed to an overload of information on short-form video applications. Therefore, a reversed 5-point Likert scale was applied. In this way, the indicators pointed in the same direction, with higher values identifying higher levels of human suffering [62]. Respondents answered with their level of agreement with the following questions: (1) “For the most part, I think life is close to ideal”; (2) “I think my living conditions are very good”; (3) “I am satisfied with my life”; (4) “So far, I have achieved the things that are important in my life”; or (5) “Even if I could start my life over again, I would not change anything” (M = 3.10, SD = 0.77, α = 0.85).

#### 3.2.5. Intention to Discontinue the Use of Short-Form Video Applications

The intention to discontinue the use of short-form video applications scale was adapted from Maier et al. [24]. This scale aims to assess individuals’ willingness to discontinue the use of short-form video applications. Respondents answered with their level of agreement with the following questions using a 4-point Likert scale: (1) “In the future, I will reduce the time I spend on short-form video applications”; (2) “I am planning to take a break from using short-form video applications for a while, but I might use them later”; (3) “I will delete my account on short-form video applications”; or (4) “I have found something more valuable than short-form video applications” (M = 2.88, SD = 0.86, α = 0.84). In order to validate the measurement scales in this study, we performed an explanatory factor analysis (EFA) to evaluate each scale’s level of adequacy. The Kaiser–Meyer–Olkin (KMO) test and Bartlett’s test of sphericity were applied to perceived information overload (0.62), short-form video fatigue (0.76), maladaptive coping (0.75), life dissatisfaction (0.85), and the intention to discontinue the use of short-form video applications (0.80). KMO values ranged between 0.6 and 1.0, which indicated that the sampling was adequate [63]. Therefore, this study’s measurement scales were suitable for the data sets. The EFA values of the measurements can be seen in Table 1.

## 4. Results

### 4.1. Descriptive Data

In total, 340 valid responses were collected. The demographic characteristics of the survey participants are outlined in Table 2. The respondents were mostly female (N = 221, 65%), single (N = 191, 56.2%), and either undergraduates (N = 219, 64.4%) or postgraduates (N = 83, 24.4). The respondents’ ages ranged from 18 to 21 years old (N = 247, 72.6%), and they had monthly incomes ranging from CNY 7000 to CNY 14,000 (N = 125, 36.8%). The bivariate associations between the key variables are provided in Table 3.

### 4.2. Hypothesis Testing

For testing Hypothesis 1, this study applied four hierarchical regression analyses for the intention to discontinue the use of short-form video applications, short-form video fatigue, maladaptive coping, and life dissatisfaction. As dependent variables, gender, education, and income were entered in the first block as controlling confounders, and perceived information overload was entered in the second block. The effect of Chinese young adults’ perceived information overload on their intention to discontinue their use of short-form video applications (β = 0.23, *p* < 0.001), short-form video fatigue (β = 0.43, *p* < 0.001), maladaptive coping (β = 0.45, *p* < 0.001), and life dissatisfaction (β = 0.30, *p* < 0.001) were all found to be significant. Therefore, H1(a–d) are fully supported (see Table 4).

Hayes’ PROCESS macro (model 4) was used to test the mediation analysis of the cognitive and psychological factors’ effects on the relationship between perceived information overload and the intention to discontinue the use of short-form video applications among Chinese young adults. The current study applied bootstrapping to obtain bias-corrected 95% confidence intervals in order to make statistical inferences about specific indirect effects. Figure 2 indicates the standardized coefficients and significance values for each path in the hypothesized model.

The results show that, in the first mediation model, short-form video fatigue bias positively predicted the intention to discontinue the use of short-form video applications (β = 0.42, *p* < 0.001). Meanwhile, perceived information overload did not predict the intention to discontinue the use of short-form video applications (β = 0.45, *p* > 0.05). The indirect effect was significant (β = 0.18, *p* < 0.001, 95% CI [0.10, 0.28]). Therefore, the full mediation effect of short-form video fatigue was confirmed. The second mediation model’s result indicated that maladaptive coping positively predicted the intention to discontinue the use of short-form video applications (β = 0.28, *p* < 0.001). Perceived information overload did not predict the intention to discontinue the use of short-form video applications (β = 0.11, *p* > 0.05). The indirect effect was significant (β = 0.12, *p* < 0.001, 95% CI [0.05, 0.20]). Thus, the full mediation effect of maladaptive coping was confirmed. The third mediation model showed that life dissatisfaction positively predicted the intention to discontinue the use of short-form video applications (β = 0.23, *p* < 0.01). Perceived information overload significantly predicted the intention to discontinue the use of short-form video applications (β = 0.16, *p* < 0.01). The indirect effect was significant (β = 0.07, *p* < 0.001, 95% CI [0.02, 0.13]) The partial mediation effect of life dissatisfaction was confirmed. Therefore, H2 and H3 were fully supported.

## 5. Discussion

Drawing on stress-coping theory, this study explains the effect of Chinese young adult users’ perceived information overload on discontinuing their use of short-form video applications. Moreover, the prime objective of this study was to examine the direct and indirect effects of cognitive (social media fatigue and maladaptive coping) and psychological (life dissatisfaction) factors on Chinese users’ intention to stop using short-form video applications.

In terms of direct effects, perceived information overload was positively associated with Chinese young adults’ discontinued use of short-form video applications. This finding is consistent with those of previous studies [10,12]. For instance, Xie and Tsai [10] indicated that there was a positive relationship between individuals’ perceived information overload and Weibo users’ discontinued usage of the platform. An overview of the past literature showed that there has been limited investigation of the effect of the perceived information overload resulting from short-form video applications on cognitive factors (such as media fatigue). However, this study anticipated that perceived information overload would be positively associated with Chinese young adult users’ short-form video fatigue. This result is in line with those of previous studies, where it was shown that individuals’ perceived excessive exposure to information on social media increased their level of media fatigue [11,27,45]. Among those studies, a previous Chinese study found that perceived information overload on social networks is positively associated with Chinese media users’ platform fatigue on these networks [45]. Stress-coping theory explained how people manage the adverse effects of stress [26]. Among the different types of coping strategies, the use of maladaptive coping strategies may increase due to perceived technostress while using social media (e.g., perceived information overload and binge-watching videos) [53,64]. This study also found that there was a positive relationship between perceived information overload and Chinese young adults’ maladaptive coping, showing continued support for this hypothesis [22,53]. Noticeably, among the negative outcomes of problematic media use, the impact on maladaptive coping was found to be significantly higher compared to the impact on other factors. Similarly, Lewis et al. [65] also found that individuals’ use of maladaptive coping is one of the most significant outcomes predicted by individuals’ frequent use of media to obtain COVID-19-related information. Lastly, this finding demonstrated that perceived information overload has a significant and positive effect on life dissatisfaction. This finding is consistent with that of a prior study showing that media users being exposed to information overload on social media amplifies their life dissatisfaction [50].

In regard to the mediated effect, firstly, short-form video fatigue mediates the relationship between perceived information overload and Chinese young adults’ discontinued use of short-form video applications. This shows consistency with previous studies, which have demonstrated that social media users’ perceived excessive exposure to COVID-19-related information has increased their level of social media fatigue. Ultimately, this has further reinforced users’ intentions to stop using social media [11,52]. Secondly, maladaptive coping acts as a mediator in the relationship between perceived information overload and Chinese young adults’ discontinued use of short-form video applications. This finding is in line with a previous study that found information overload on social media to be a predictor of individuals’ level of maladaptive coping, which in turn further reinforces their intention to discontinue their use of social media [22]. Lastly, Cleofas [66] indicated that college students tended to use social media (e.g., Facebook, Twitter, Instagram, YouTube, and TikTok) to cope during the COVID-19 lockdown period. Additionally, this study also demonstrated that the greater an individual’s daily use of TikTok, the more likely it was to increase their social media disorder. Eventually, this may increase individuals’ levels of depression, loneliness, low self-esteem, poor sleep quality, etc. [67]. Such negative feelings are also known as “life dissatisfaction”. A prior study demonstrated that life dissatisfaction is positively associated with individuals avoiding the use of social media [24]. The current study’s finding is consistent with those of previous studies [24,25,50], e.g., that life dissatisfaction mediates the relationship between perceived information overload and Chinese young adults’ discontinued use of short-form video applications.

The practical implications of this study relate to three aspects. Firstly, even though short-form video applications have brought a lot of benefits to Chinese people amid the COVID-19 pandemic, they have led to negative effects that are amplified by information overload. Therefore, the providers of short-form video applications should be aware of the potential negative outcomes. Additionally, reducing the number of daily notifications and instead setting more user-friendly alerts, such as push notifications, should ensure that users receive a limited amount of daily information. Secondly, the current study provides empirical evidence that short-form video fatigue is one of the key mediators in the relationship between perceived information overload and the discontinued use of short-form video applications. Moreover, spending time on these applications may induce individuals to experience short-form video fatigue, resulting in users quitting short-form video applications. Therefore, these applications’ providers should also pay more attention to the users’ time limitation when using these applications. In addition, the applications’ service providers can add features to applications that remind users of the recommended healthy amounts of time to spend on these applications. Social support has been shown to be one of the key factors that can offset the level of depression (life dissatisfaction) [68]. Therefore, encouraging family members and work colleagues to share helpful suggestions and guide each other in limiting their exposure to information on short-form video applications can help them manage their wellbeing.

There are several limitations of this study. Firstly, the analysis was conducted using a cross-sectional study. In this type of investigation, it is difficult to identify the direct and indirect effects of the tested factors. Secondly, the respondents’ education was mainly at the undergraduate level. Therefore, these findings cannot be generalized to young adult users in China from all education groups. Lastly, the current study adapted the original scales for measuring individuals discontinuing the use of short-form video applications. Undeniably, it is difficult to measure their actual behavior change. Instead, this study used the perspective of “expected” or “perceived” as a behavior change. A more accurate measurement would be beneficial for future studies.

Future research needs to further investigate the impacts of cognitive and psychological factors. For instance, whether self-efficacy is directly or indirectly associated with the discontinued use of short-form video applications should be analyzed. Similarly, whether an individual’s social comparison is positively or negatively associated with users avoiding the use of short-form video should also be investigated. These potential future studies will contribute to providing a holistic perspective of short-form video users’ behavior.

## 6. Conclusions

The current study discovered how information overload affects Chinese young adult users’ (18–26 years old) intention to discontinue using short-form video applications during the COVID-19 pandemic in China. This is the first study to explore both the direct and indirect effects of critical factors (short-form video fatigue, maladaptive coping, and life dissatisfaction) on such a relationship. The findings of this study make contributions to behavioral science research in the following aspects: First, the current study fills a gap in the prior literature on the discontinued use of short-form video applications in the context of the COVID-19 pandemic in China. Second, this study confirms that both cognitive (social media fatigue and maladaptive coping) and psychological (life dissatisfaction) factors act as mediating mechanisms in the relationship between information overload and Chinese young adult users’ intention to discontinue using short-form video applications. Future studies are recommended to investigate whether self-efficacy and social comparison are directly or indirectly associated with discontinuing the use of short-form video applications. The exploration of these factors will provide a better understanding of the negative outcomes of Chinese short-form video application users’ problematic media use.

## Figures and Tables

**Figure 1 behavsci-13-00050-f001:**
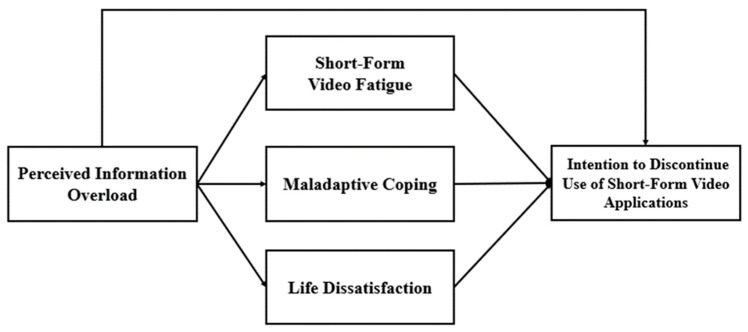
A model of the predictors of the intention to discontinue the use of short-form video applications.

**Figure 2 behavsci-13-00050-f002:**
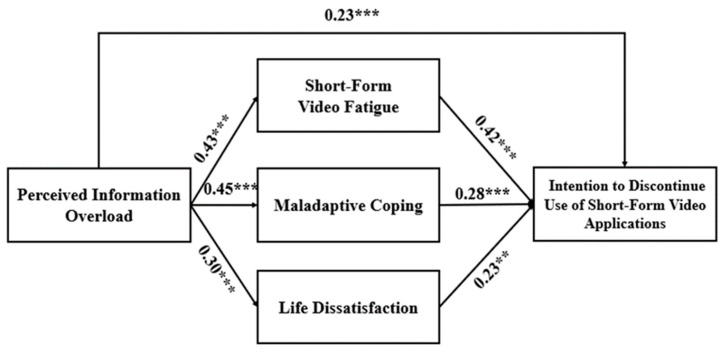
The effects of predictors of the intention to discontinue the use of short-form video applications. ** *p* < 0.01, *** *p* < 0.001.

**Table 1 behavsci-13-00050-t001:** CR, CA, and AVE values.

Variables	CR	CA	AVE
Perceived Information Overload	0.77	0.65	0.54
Short-Form Video Fatigue	0.85	0.79	0.60
Maladaptive Coping	0.90	0.84	0.67
Life Dissatisfaction	0.90	0.85	0.64
Intention to Discontinue Use of Short-Form Video Applications	0.90	0.84	0.66

CA: Cronbach’s-alpha; CR: composite reliability; AVE: average variance extracted.

**Table 2 behavsci-13-00050-t002:** Key demographic characteristics of the survey participants.

Variables	Item	Count	Percentage
Gender	Female	221	65.0%
	Male	119	35.0%
Education Level	High school	9	2.6%
	Pre-college	10	2.9%
	Undergraduate	219	64.4%
	Master’s	83	24.4%
	Ph.D.	19	5.6%
Age	18–21 years old	247	72.6%
	22–26 years old	93	27.4%
Marital status	Single	191	56.2%
	In relationship	93	27.4%
	Married	56	16.5%
Monthly Income	CNY 1000–6999	115	33.8%
	CNY 7000–14,000	125	36.8%
	CNY 14,000–49,999	71	20.9%
	>CNY 50,000	29	8.5%
	Total	340	100%

**Table 3 behavsci-13-00050-t003:** Correlations between key variables.

Variables	1	2	3	4	5
Perceived Information Overload	–				
Short-form Video Fatigue	0.41 **	–			
Maladaptive Coping	0.48 **	0.28 **	–		
Life Dissatisfaction	0.32 **	0.17 **	0.34 **	–	
Intention to Discontinue Use of Short-Form Video Applications	0.28 **	0.47 **	0.31 **	0.26 **	–
Mean	2.80	2.94	2.98	3.10	2.88
SD	0.85	0.91	0.78	0.77	0.86
α	0.65	0.84	0.79	0.85	0.84

** *p* < 0.01.

**Table 4 behavsci-13-00050-t004:** Regression analysis predictors of cognitive and psychological factors.

	IDUSVA	SFVF	MC	LD
Gender	0.03	−0.03	−0.02	0.15
Education	−0.02	−0.02	0.04	0.04
Income	0.02	0.52 *	0.06	0.03
Adj.R^2^	−0.05	0.11	−0.08	0.12
R^2^ change	0.01	0.02	0.01	0.02
PIO	0.23 **	0.43 **	0.45 ***	0.30 ***
Adj.R^2^	0.42 ***	0.17 ***	0.23 ***	0.11 ***
R^2^ change	0.05 **	0.16 ***	0.27 ***	0.10 ***

* *p* < 0.05, ** *p* < 0.01, *** *p* < 0.001; PIO = Perceived Information Overload. IDUSVA = Intention to Discontinue Use of Short-Form Video Applications. SFVF = Short-form Video Fatigue. MC = Maladaptive Coping. LD = Life Dissatisfaction.

## Data Availability

Not applicable.

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
