# Peer review of "Perceived Information Overload and Intention to Discontinue Use of Short-Form Video: The Mediating Roles of Cognitive and Psychological Factors"

_behavsci, 2023, doi:10.3390/bs13010050_

Round 1
Reviewer 1 Report
The manuscript is an original scientific paper that focuses on influence of information overload on the intention to discontinue using short-form video apps and mediating effects of selected cognitive and psychological factors. Information overload is an important concept, especially in the light of the covid-19 pandemic where social and communication practices migrated to social media platform even to a greater extent than before. The topic is therefore relevant and interesting from scientific point of view. The authors have made a good work by preparing the manuscript. The core strength of the manuscript is the proposed model, however, there are several inconsistencies in some parts, as well as introduction, results and discussion sections need to be improved. I suggest to deep edit the manuscript and to proofread it by native English proofreader. Other suggestions for improvement are listed in the comments to authors section.
GENERAL:
· The manuscript needs to be deep edited and proofread by a native English speaker.
ABSTRACT:
INTRODUCTION:
· line 27: In recent years, social media influence has improved continuously – Did you mean has developed/has changed/has evolved?
· General comment: please revise the introduction part, write it more concisely, use more references to back up the information provided in the introduction. Please introduce/define the term short-form video more into detail Elaborate more more into details why discontinued use is important. Elaborate more into details why did you choose selected cognitive and psychological factors? Introduce stress-coping theory to the reader. Why did you use this framework? Derive the aim of the manuscript more precisely and more into details.
· lines 94-96: please reword the sentence – it is not fully understandable what is ment.
· line 98: Information Overload on Intention, Cognitive and Psychological Factors – is there something missing? Influence of information overload maybe? Or effect? Or something else?
· paragraph titled 2.2. Information Overload on Intention, Cognitive and Psychological Factors – define short-form video applications. And elaborate on why it is important to know the dynamics behind the effects of perceived information overload on users' intention to stop using short-form video applications. Moreover, deep edit the text and write it down more conceisly, more focused. It needs clearance.
· lines 187-190: H2a, H2b and H3 need rewording and proofreading. It is not completely clear what is ment. Please clarify the relations. I assume it is ment that for example for H3: Life dissatisfaction mediates the relationship between perceived information overload AND intention to discontinue use of short-form video applications. I assume the same goes for H2a and H2b as well. Please be more precise.
METHODS:
· Please provide more information on the methodology (type of study, type of assessment). Please describe more used questionnaire concisely – were any other parts beside the existing (psychological) assessment tools/scales you used to measure your main concepts (perceived information overload, intention to discontinue use of short-form video apps, short-form video fatigue, maladaptive coping, life dissatisfaction)? Had they been previously thoroughly validated for the use within Chinese cultural context?
RESULTS:
· line 261 – Descriptive data – I assume this is already a part of the results – is the title Results or any other part of the manuscript missing here?
DISCUSSION:
· Please take into account all the proposed changes I mentioned within the introduction part and revise the discussion part. Add some new information next to previous studies being mentioned. Add some more information that will support the findings. Mind adding some new perspective on the content presented in the manuscript.
CONCLUSIONS:
Add conclusions. Try to draw conclusions beyond what you have already stated in the discussion section.
Reviewer 2 Report
The topic of the article is significant and contemporary from both an individual and societal standpoint. The outlined divide into a section presenting relevant literature and a part focused on the research strikes me as a positive aspect of the article. Sound knowledge of the literature and its appropriate selection are demonstrated in the first section. The second half, however, is overly constrained to the indexing of the research findings and lacks a comprehensive analysis of these findings.
The article's quality has suffered since some topics were handled in a chaotic and understated manner. I'm referring here to the frequent mentions of the Covid-19 pandemic in the text that are not preceded by a clear indication of whether the conducted study is explicitly intended to examine how the pandemic has affected the intensification of information overload or whether the pandemic occurs, as it were, "by accident." The function the pandemic serves in this context is unclear. This element calls for clarification. However, if the pandemic aspect is indeed significant, perhaps it should be mentioned in the title.
One should bear in mind that the pandemic situation caused by SARS-COV-2 (COVID-19) has had a major impact upon the sense of health hazard, which combined with economic uncertainly and forced social isolation, has had harmful effects for individuals. Studies show that current mental health issues like anxiety or depression grow notably worse during certain dramatic scenarios, such as wars or economic crises. Thus, the sheer information overload in question may show to be one of the factors, but not the only one, causing some of those interviewed to stop using short video applications.
The overuse of some vague terms and phrases is, in my opinion, less significant but nonetheless noteworthy, such as: „Previous research” (34,36,154,180), „A previous Chinese scholar” (141), Previously (161), „A handful of researchers” (173), previous literature (199), etc, etc. It is challenging to determine which studies the authors are referencing in the text.
What’s more, it has not been clearly stated why the authors recall South Korean and Israeli studies, what motivated them to choose these countries.
I am convinced that because of how important the subject matter is and how fascinating the research findings are, the article deserves to be published, yet significant revisions must still be made.
Round 2
Reviewer 1 Report
Dear authors.
Thank you for providing edited manuscript. You did a good job with edits made. In its present form now the manuscript is way clearer, precise and coherent.